# Unraveling the Metabolic Pathways Between Metabolic-Associated Fatty Liver Disease (MAFLD) and Sarcopenia

**DOI:** 10.3390/ijms26104673

**Published:** 2025-05-14

**Authors:** Marina Ribas Losasso, Maria Luiza Cesto Parussolo, Antony Oliveira Silva, Rosa Direito, Karina Quesada, Claudia Rucco Penteado Detregiachi, Marcelo Dib Bechara, Nahum Méndez-Sánchez, Ludovico Abenavoli, Adriano Cressoni Araújo, Ricardo de Alvares Goulart, Elen Landgraf Guiger, Lucas Fornari Laurindo, Sandra Maria Barbalho

**Affiliations:** 1Department of Biochemistry and Pharmacology, School of Medicine, Universidade de Marília (UNIMAR), Marília 17525-902, SP, Brazil; 2Laboratory of Systems Integration Pharmacology, Clinical and Regulatory Science, Research Institute for Medicines, Universidade de Lisboa (iMed.ULisboa), Av. Prof. Gama Pinto, 1649-003 Lisbon, Portugal; 3Postgraduate Program in Structural and Functional Interactions in Rehabilitation, School of Medicine, Universidade de Marília (UNIMAR), Marília 17525-902, SP, Brazil; 4Liver Research Unit, Medica Sur Clinic & Foundation, Mexico City 14050, Mexico; 5Faculty of Medicine, National Autonomous University of Mexico, Mexico City 04510, Mexico; 6Department of Health Sciences, University “Magna Graecia”, Viale Europa, 88100 Catanzaro, Italy; 7Department of Biochemistry and Nutrition, School of Food and Technology of Marília (FATEC), Marília 17500-000, SP, Brazil; 8Research Coordinator, UNIMAR Charity Hospital, Universidade de Marília (UNIMAR), Marília 17525-902, SP, Brazil

**Keywords:** metabolic-associated fatty liver disease, MAFLD, liver disease, sarcopenia, cardiovascular diseases, metabolic diseases

## Abstract

Metabolic-Associated Fatty Liver Disease (MAFLD) is a public health concern that is constantly expanding, with a fast-growing prevalence, and it affects about a quarter of the world’s population. This condition is a significant risk factor for cardiovascular, hepatic, and oncologic diseases, such as hypertension, hepatoma, and atherosclerosis. Sarcopenia was long considered to be an aging-related syndrome, but today, it is acknowledged to be secondarily related to chronic diseases such as metabolic syndrome, cardiovascular conditions, and liver diseases, among other comorbidities associated with insulin resistance and chronic inflammation, besides inactivity and poor nutrition. The physiopathology involving MAFLD and sarcopenia has still not been solved. Inflammation, oxidative stress, mitochondrial dysfunction, and insulin resistance seem to be some of the keys to this relationship since this hormone target is mainly the skeletal muscle. This review aimed to comprehensively discuss the main metabolic and physiological pathways involved in these conditions. MAFLD and sarcopenia are interconnected by a complex network of pathophysiological mechanisms, such as insulin resistance, skeletal muscle tissue production capacity, chronic inflammatory state, oxidative stress, and mitochondrial dysfunction, which are the main contributors to this relationship. In addition, in a clinical analysis, patients with sarcopenia and MAFLD manifest more severe hepatitis fibrosis when compared to patients with only MAFLD. These patients, with both disorders, also present clinical improvement in their MAFLD when treated for sarcopenia, reinforcing the association between them. Lifestyle changes accompanied by non-pharmacological interventions, such as dietary therapy and increased physical activity, undoubtedly improve this scenario.

## 1. Introduction

Fatty liver disease is a public health concern that is constantly expanding, with a fast-growing prevalence, and it affects about a quarter of the world’s population. This condition is a significant risk factor for cardiovascular, hepatic, and oncologic diseases, such as hypertension, hepatoma, and atherosclerosis. Therefore, it is globally one of the most common causes of mortality [1,2,3,4,5].

In 2020, the International Consensus Panel defined new terminology for the liver disease spectrum that embraces fatty liver, systemic metabolic dysfunctions, and potential comorbidities, denominated metabolic dysfunction-associated fatty liver disease (MAFLD). This new terminology replaces the acronym NAFLD, which was used to delimit the fatty liver unrelated to alcohol consumption or other agents that lead to hepatic damage. The major cause of this alteration was the addition of metabolic dysfunction as a criterion [1,6,7,8,9,10]. The term NAFLD was replaced by MAFLD to more broadly designate a spectrum of metabolic diseases, and these new criteria expand early diagnoses across the population while promoting the early establishment of a specific and effective therapeutic regimen. In addition, one of the main aspects that this terminology change implies is the removal of secondary causes of liver disease from the position of diagnostic exclusion criteria, which allows for the amplified treatment of liver diseases. Moreover, experts have increasingly discussed the theory that this change will contribute to increased understanding in society about MAFLD, since only about 4% of the population reported knowing the old, restricted definition of NAFLD [11,12,13,14].

MAFLD is one of the most common liver diseases in the world and significantly impacts the world’s population [5,6,9,15]. The diagnosis involves hepatic steatosis associated with one of the three following parameters: type 2 diabetes; overweight or obesity; or average weight with not less than two metabolic dysfunctions, like prediabetes, high blood pressure, and dyslipidemia [8,16,17]. Those criteria have punctuated the contribution of metabolic dysregulation in the process of liver dysfunction; all of these risk circumstances are directly related to liver fat deposition or associated with the disease progression and genesis of hepatic or extrahepatic complications [11,18]. One example is insulin resistance in the muscle and adipose tissue, which affects hepatic lipid metabolism, thereby promoting hepatic fat deposition and inflammation [9,19,20]. Figure 1 shows factors that may influence the pathogenesis of MAFLD in a patient with metabolic syndrome.

Sarcopenia is a condition associated with deficient muscle strength and low muscle mass or quality and physical performance. Poor muscle strength is the major parameter to consider in this disease, while the presence of the other criteria in parallel validates the diagnosis and indicates the severity of this condition [21,22,23,24]. To the diagnosis, these qualifications are identified by methods to measure skeletal muscle mass, which are commonly bioelectrical impedance analysis (BIA) or dual-energy X-ray absorptiometry (DEXA) [25]. Along with these techniques, the European Working Group on Sarcopenia in Older People 2 (EWGSOP2) suggests that the SARC-F (acronym for “Strength, Assistance with walking, Rising from chair, Climbing stairs, and Falls”) questionnaire succeeds as a screening for sarcopenia. Other effective methods include the grip strength test [26,27]. Sarcopenia was long considered to be an aging-related syndrome, but today, it is acknowledged to be secondarily related to chronic diseases such as metabolic syndrome, cardiovascular conditions, and liver diseases, among other comorbidities associated with insulin resistance and chronic inflammation, besides inactivity and poor nutrition [28,29,30,31]. Among those chronic diseases, MAFLD is reported to be connected with sarcopenia, as both share multiple pathogenic mechanisms, such as insulin resistance and cytokine release imbalance [32,33,34]. Figure 2 highlights the differences between a healthy and a diseased muscle with sarcopenia and illustrates the main points of the pathology.

The physiopathology involving MAFLD and sarcopenia has still not been solved. Insulin resistance, a component of MAFLD, might be one of the keys to this relationship since this hormone target is mainly the skeletal muscle. The levels of inflammatory markers such as TNF-α, high-sensitivity C-reactive protein (hs-CRP), and interleukin (IL)-6 are above average in patients with lower muscle strength, suggesting that sarcopenia might have a relationship with MAFLD [35,36,37,38]. TNF-α, for example, is present in the physiopathology of MAFLD due to the activation of pro-inflammatory macrophages during the pathological hypertrophy of adipocytes, promoting lipolysis and contributing to insulin resistance by acting on the insulin receptor and inducing the phosphorylation of Ser307 instead of tyrosine on Irs 1 [19,39]. The insulin resistance caused by the higher levels of cytokines can also cause muscle degradation [40,41].

Aside from insulin resistance, there are other shared mechanisms, such as the secretory role of skeletal muscle. The release of myokines acts in the balance of glucose and fatty acid metabolism. Deficient muscle mass can disturb the levels of myokines, such as IL-6, myostatin, and adiponectin, thereby overcoming fat accumulation in the liver. On the other hand, patients with liver disease can also contribute to the alteration of myokine levels, exposing the skeletal muscle to a high ammonia concentration, a result of hepatocyte dysfunction, a damaged urea cycle, and the dysregulation of the intestinal microbiota [38,42]. Moreover, serum levels of fibroblast growth factor-21 (FGF-21) are much higher in decompensated cirrhosis patients with sarcopenia. FGF21 works as a liver–muscle crosstalk, thereby disrupting muscle regeneration due to inhibiting the PI3K/Akt pathways [38]. Finally, one of the risk factors for developing MAFDL is sedentarism, which is known to be related to muscle strength deficits [43,44]. In sum, MAFLD and sarcopenia embrace shared mechanisms of physiopathology, such as oxidative stress, chronic inflammation, and insulin resistance; in that manner, the presence of one disease contributes to the development of the other. Apart from the physiopathological relationship, it is important to understand the clinical connection between MAFLD and sarcopenia to recognize high-risk patients and perform adequate clinical management. Moreover, the early identification of the relationship between those conditions can improve the patient’s prognosis. Among the shared risk factors that contribute to metabolic dysregulation are metabolic syndrome, hypercholesterolemia, the male gender, diabetes mellitus (elevated glycated hemoglobin levels), and alcoholism [6,35,45].

In a recent meta-analysis, some authors have shown that sarcopenia increases the risk for MAFLD and is associated with an increased risk of liver fibrosis in these patients [46]. Although this study was very interesting and showed important results, the authors did not outline the metabolic processes that may relate to the two conditions.

In view of the important relationship between liver conditions and sarcopenia, this review aims to comprehensively discuss the main metabolic and physiological pathways involved in these conditions.

## 2. Literature Search

The articles included in this review were captured from the PUBMED, EMBASE, and Cochrane databases. The terms used were Non-alcoholic fatty liver disease or Metabolic-associated fatty liver disease or MAFLD or non-alcoholic fatty liver disease or NAFLD or non-alcoholic steatohepatitis or NASH and sarcopenia or muscle strength or muscle mass or muscle proteolysis. We only included articles published in the English language. Reviews, clinical trials, and meta-analyses were included. Poster presentations and grey literature were excluded from our study.

## 3. MAFLD, Inflammatory Processes, Oxidative Stress, and Mitochondrial Dysfunction

MAFLD presents a heterogeneous pathophysiology initially described through the “two-hit hypothesis”, which considers hepatic steatosis as the first significant damage induced by insulin resistance, while oxidative stress represents the second hit, exacerbating inflammation. Fibrosis is also affected in the long term due to this process. This hypothesis has been replaced by a more complex, multifactorial model known as “multiple hits”, in which various factors act simultaneously. Chronic inflammation, along with oxidative stress, contributes to insulin resistance. Pro-inflammatory mediators such as leptin, resistin, IL-6, TNF-α, and intestinal bacterial endotoxins interfere with insulin signaling, impairing glucose uptake in peripheral tissues [47,48,49]. Additionally, oxidative stress resulting from the accumulation of fatty acids in the liver contributes to mitochondrial dysfunction and the activation of pro-inflammatory signaling pathways, thereby increasing insulin resistance. Chronic inflammation is crucial for the progression of MAFLD, with the activation of inflammatory pathways and an imbalance of pro-inflammatory and anti-inflammatory adipokines contributing to the transition from simple hepatic steatosis to more severe forms of the disease [50,51,52,53]. Moreover, genetic polymorphisms in the PNPLA3 and TM6SF2 genes are linked to the development of MAFLD. These variants positively favor the accumulation of glycerides in the hepatocyte and increase the likelihood of fibrosis severity. While polymorphisms in PNPLA3 are related to increased lipids, TM6SF2 E167K is related to the reduced secretion of VLDL, leading to greater fat accumulation [54].

Oxidative stress is one of the most prominent features of this new model and involves a disproportion between the production and elimination of reactive oxygen species (ROS). Normal cellular metabolism produces a certain amount of ROS through processes such as cellular respiration and β-oxidation; however, cellular antioxidant activity, through enzymes such as catalase, superoxide dismutase, and glutathione peroxidase—as well as polyphenols and mitochondrial mechanisms—can counteract this production, maintaining biological balance. A high-fat diet, however, increases the substrate available for oxidative processes, deteriorates antioxidant mechanisms, and enhances the production of free radicals [55]. For instance, increased glucose intake results in the higher production of pyruvate, the initial substrate for β-oxidation, which generates acetyl-CoA and ROS. The accumulation of ROS induces the oxidation of nucleic acids, proteins, and lipids, thereby compromising cellular function and leading to the production of pro-inflammatory cytokines such as TNF-α, IL-6, IL-1β, and Transforming Growth Factor-β (TGF-β) [56,57,58]. Transcription factors such as nuclear factor kappa B (NF-κB) and mitogen-activated protein kinase (MAPK) are activated, regulating the expression of inflammatory genes [59]. Simultaneously, toll-like receptors (TLRs) are expressed in hepatocytes and non-parenchymal cells in the region, leading to the release of pro-inflammatory cytokines and chemokines, which, along with interleukins and prostaglandins, attract immune cells to the site of inflammation [60,61,62,63,64]. TNF-α induces the production of additional cytokines and promotes the migration and activation of defense cells to the inflammation site. TGF-β drives fibroblast proliferation, leading to fibrosis and reduced hepatic function. Additionally, insulin resistance can further increase the production of advanced glycation end-products (AGEs), which exacerbate oxidative stress and pro-inflammatory pathways. This relationship thus contributes to oxidative stress, thereby creating a biological imbalance that results in initial cellular changes evolving into systemic manifestations [65,66,67,68,69].

Kupffer cells are an example of cells that are directly affected by the overproduction of ROS since only exposure to them generates cell activation, which in turn contributes to the production of ROS. In addition, a stimulus for the production of cytokines and chemokines favors the installation of an inflammatory process responsible for mediating processes of cell damage and fibrogenesis that also extend to hepatocytes. Moreover, when hepatic stellate cells are activated, also by the production of ROS, they increase the production of extracellular matrix components, evolving into fibrosis and, in more advanced cases, cirrhosis [65,70,71].

The presence of MAFLD can also be related to mitochondrial dysfunction. In this case, there is an increased mitochondrial permeability, leading to the loss of calcium ions and diminishing the concentration of protons that would feed the electron transport chain. This scenario results in insufficient energy (ATP) production and an augmentation of calcium cytoplasmic concentration. This imbalance in the gradient concentration inside and outside the mitochondria can result in structural alterations and even organelle destruction [72,73,74,75,76,77,78]. Free radicals may promote this impairment in mitochondrial structure and function and can result in the increased production of these reactive species [79,80]. Many consequences can be observed, including lipotoxic accumulation associated with MAFLD [81,82,83,84,85]. In hepatic steatosis, there is an increase in mitochondrial fission and decreased fusion, which could be normal in healthy conditions [86,87]. However, the fission process, affected by increased levels of reactive species, accelerates mitochondrial DNA fragmentation and increases the production of ROS, contributing to MAFLD evolution. This notwithstanding, researchers have shown that new mitochondria formed under these conditions may be defective [72,88,89,90,91,92,93,94,95].

Another repercussion of mitochondrial dysfunction is nutrition overload, which accelerates fatty acid oxidation through the Krebs cycle, leading to ROS overproduction. This large amount of ROS causes irreversible damage to the mitochondrial electron transfer chain, leading to mitochondrial dysfunction, inflammation, cellular apoptosis, and liver fibrosis [96,97,98,99].

The mitophagy pathways are beneficial for removing problematic mitochondria and excessive oxidative toxic products. In patients with MAFLD, this process is inhibited, leading to increased concentrations of reactive species inside the cell, resulting in apoptosis and worsening oxidative stress [100,101,102,103,104].

In MAFLD, dysfunctional mitochondrial metabolism and excessive ROS production are common. Also, this scenario affects the endoplasmic reticulum, which also has a role in metabolite exchange through complex polymeric protein structures [105,106,107,108,109]. In the imbalance of ER homeostasis or insufficient energy, the endoplasmic reticulum is stimulated, and there is an imbalance in the distribution of glutathione and oxidized glutathione, inducing mitochondrial stress and impaired mitochondrial ROS production, thereby aggravating oxidative stress [110,111].

As seen above, a close relationship exists between mitochondrial function, oxidative stress, and MAFLD. These conditions are closely linked to inflammatory processes, which collectively contribute to its pathogenesis and progression.

## 4. Sarcopenia

Sarcopenia is a progressive and widespread disorder that affects skeletal muscles, leading to the loss of muscle mass, strength, and function [112,113,114,115,116]. This condition is commonly associated with aging and chronic health issues such as inflammation, insulin resistance, oxidative stress, mitochondrial dysfunctions, and neuromuscular junction dysfunction [117,118]. Sarcopenia has harmful health effects, increasing morbidity and mortality, but these effects can be prevented with the early identification of risk factors and lifestyle changes [31]. The prevalence of sarcopenia ranges from 10% to 16% of older individuals worldwide, 18% of diabetic individuals, and approximately 66% of unresectable esophageal malignancy cases [119].

Despite advances, there is still no international consensus on the definition and diagnosis of sarcopenia [120,121]. The most well-established definitions are based on the Sarcopenia Definitions and Outcomes Consortium (SDOC), the Asian definition, and the European Working Group on Sarcopenia in Older People (EWGSOP), which are used globally. These definitions and diagnostic criteria have improved clinical management, the search for effective treatments, epidemiology, and the understanding of pathophysiological mechanisms [122].

Currently, treatment is divided into pharmacological and non-pharmacological approaches. The effectiveness of pharmacological treatment, which includes steroids, β-receptor blockers, troponin activators, growth hormone, angiotensin-converting enzyme inhibitors, selective androgen receptor modulators (SARMs), angiotensin II receptor drugs, and appetite stimulants, is still in question [123,124]. On the other hand, non-pharmacological treatment is more established; resistance exercise and optimizing one’s diet with vitamin D and protein supplementation are the most effective approaches for managing sarcopenia [125,126,127].

### Sarcopenia, Oxidative, and Inflammation Processes

The pathogenesis of sarcopenia, although still debated, is predominantly attributed to chronic low-grade inflammation associated with aging, commonly referred to as “inflammaging”. Aging leads to a decline in adaptive immune response while sustaining the chronic activation of innate immunity, resulting in tissue degeneration [128,129]. Several theories attempt to elucidate this phenomenon, including mutations caused by telomere and mitochondrial DNA damage alongside oxidative stress [130]. These processes disrupt skeletal muscle homeostasis during aging, potentially amplifying catabolic pathways and diminishing anabolic processes crucial for protein synthesis [131]. A decline in satellite cells occurs in sarcopenic muscle, alongside increased fat infiltration and reduced muscle fibers, notably type II fibers [132]. These cells activate upon adjacent muscle fiber damage, initiating repair and regeneration processes regulated by systemic factors governing their activity and differentiation. The dysfunction of the neuromuscular junction, inflammation, insulin resistance, and oxidative stress contribute significantly to muscle loss [133,134].

“Inflammaging” is characterized by heightened serum levels of pro-inflammatory cytokines such as tumor necrosis factor-alpha (TNF-α), interleukin (IL)-1β, IL-6, IL-18, and CRP, the latter synthesized in the liver under IL-6 influence. These cytokines impair mitochondrial function, reduce ATP production, and generate ROS [135,136]. Excess ROS further compromises mitochondrial integrity and stimulates proteolysis via heightened ubiquitin–proteasome system activity, ultimately promoting muscle atrophy [137,138]. Elevated TNF-α levels induce excessive apoptosis in skeletal muscle by regulating apoptotic signaling through caspase activation and proteolytic mechanisms [139,140]. Obesity also exacerbates chronic inflammation through adipose tissue interactions with immune cells like macrophages and neutrophils. Macrophage-secreted TNF-α promotes adipose tissue lipolysis via the TLR4 pathway, elevating free fatty acids and triggering the release of pro-inflammatory cytokines such as IL-1β and IL-6 in response to increased free fatty acids [132,133,141]. Interleukins exert a pivotal role in sarcopenia, with elevated TNF-α, IL-6, and IL-1β levels correlating with increased mortality and morbidity among older adults [135]. IL-6, a myokine, functions prominently during the acute inflammatory response and metabolic regulation. Ongoing research suggests IL-6′s dual role in limiting inflammation by inhibiting TNF-α and IL-1β production, thereby regulating neutrophil recruitment. IL-6 secretion increases with physical activity, underscoring its production in muscle tissue. Studies also associate elevated IL-1β levels with sarcopenia, highlighting its role in mediating inflammatory responses and influencing cellular proliferation, differentiation, and apoptosis, primarily orchestrated by macrophages [142,143,144].

In addition, when examining the pathophysiology of sarcopenia, it is important to understand the association between mitochondrial dysfunction and the inflammatory processes previously elucidated. In muscle homeostasis, mitochondria play several roles, such as the self-digestion of internal components, the control of programmed cell death, and the production of ROS, which can also be added to functions related to protein control. Thus, mitochondrial dysfunction corresponds to an important mechanism of skeletal muscle degradation and the development of sarcopenia [145,146,147,148].

For a satisfactory mitochondrial activity, it is necessary for a metabolic arrangement between several physiological processes that, in turn, are catalyzed by several enzymes, such as pyruvate dehydrogenase complex (PDC), which is regulated by pyruvate dehydrogenase kinases (PDKs), a second group of mitochondrial enzymes, with emphasis on PDK4. When analyzing the cellular activity of PDK4, it is observed that controlling the activity of PDC by inhibitory phosphorylation alters the mitochondrial carbon substrate, causing disorders such as sarcopenia and insulin resistance [149]. In addition, PDK4 also mediates protease Lon peptidase 1 (LONP1), whose function is to perform a kind of mitochondrial audit and induce the removal of inappropriate ones; so, changes in PDK4 lead to a signaling cascade, resulting in skeletal muscle proteolysis [150].

Sarcopenia is related to the imbalance of multiple pathways, including oxidative stress, mitochondrial dysfunction, insulin resistance, and protein metabolism. A close look at these variables can help with preventive measures and also with the treatment of these conditions.

## 5. Role of Insulin Resistance, MAFLD, and the Relationship with Sarcopenia

Upon binding to its receptor, insulin induces an increase in tyrosine kinase activity, followed by the phosphorylation of tyrosine residues on the insulin receptor substrate (IRS). Once phosphorylated by tyrosine, IRS binds to phosphatidylinositol 3-kinase (PI3K) and activates it, leading to the formation of phosphatidylinositol 3,4,5-trisphosphate (PIP3) at the plasma membrane. The 3′-phosphoinositide-dependent kinase 1 (PDK1) is then activated by binding its pleckstrin homology (PH) domain to PIP3, which in turn activates the protein kinase AKT through the phosphorylation of a threonine residue. AKT subsequently targets various molecules, including forkhead box O1 (FoxO1), mechanistic target of rapamycin (mTOR), and glucose transporter 4 (GLUT4). Insulin resistance refers to the diminished sensitivity of target tissues to insulin stimulation. Several factors contribute to insulin resistance, including defects in the insulin signaling cascade, the inhibition of the cascade through pro-inflammatory pathways, and even insulin resistance induced by fatty acids due to lipid accumulation in the liver and muscles [19,151,152,153].

Insulin resistance plays a crucial role in the pathogenesis of MAFLD. As previously mentioned, the currently accepted model, the “two-hit hypothesis”, involves insulin resistance as the first event. This initial event is the primary factor responsible for hepatocyte lipid accumulation, leading to hepatic steatosis. This initial event renders the liver more vulnerable to the second event, which results in hepatic damage, inflammation, and fibrosis. In the liver, lipids and metabolites, such as lipoproteins and acyl-carnitines, are secreted and can interfere with insulin action as regulators. Increased levels of free fatty acids (FFAs), the inflammation of adipose tissue, and decreased adiponectin play major roles in the development of insulin resistance in MAFLD. In this condition, elevated levels of FFAs are converted into triglycerides via the glycerol-3-phosphate pathway and into ceramides and diacylglycerols (DAGs). DAGs are involved in the activation of protein kinase C (PKC), which may inhibit the insulin receptor via threonine 1160, an amino acid involved in reducing insulin resistance. Furthermore, pro-inflammatory cytokines such as IL-6 and TNF-α also contribute to insulin resistance. In obesity, a diagnostic criterion for MAFLD, there is an increase in lipid accumulation and the secretion of pro-inflammatory cytokines by adipocytes and macrophages, such as monocyte chemoattractant protein-1 (MCP-1), TNF-α, and interleukins, as well as an increase in the release of C-reactive protein (CRP), which contributes to insulin resistance through various mechanisms. These mechanisms include the activation of Ser/Thr kinases; the decreased expression of IRS-1, GLUT4, and peroxisome proliferator-activated receptor gamma γ (PPARγ); or the activation of SOCS3 in adipocytes. CRP binds to leptin, blocking leptin signaling and thus impairing the peptide’s role in regulating food intake and energy expenditure. Regarding adiponectin, it can promote the β-oxidation of fatty acids, glucose utilization, and the suppression of fatty acid synthesis, and its levels are decreased in patients with MAFLD, thus playing a role in the development of insulin resistance [56,154,155,156].

Skeletal muscle is insulin’s primary target organ and regulates glucose through its uptake and metabolism. Insulin resistance plays a crucial role in the pathogenesis of sarcopenia by inducing muscle atrophy through mechanisms such as increased protein catabolism, decreased protein synthesis, the increased expression of the FoxO family, and autophagy in skeletal muscle cells. Inflammation and dyslipidemia are diagnostic criteria for MAFLD, which induces lipolysis and leads to ectopic fat infiltration in various organs. Excess lipids tend to accumulate primarily in skeletal muscle, resulting in myosteatosis. Myocyte lipid infiltration activates pro-inflammatory pathways, generating cytokines such as TNF-α, MCP-1, and IL-6. MCP-1 causes the chemotaxis of monocytes to the site of inflammation, where they become M1 macrophages upon attaching to adipocytes. These M1 macrophages release various inflammatory factors, thereby recruiting more monocytes. TNF-α induces the activity of proteins like inhibitors of nuclear factor-κB (IκB) kinase (IKK), which directly affect the IRS of the insulin receptor and can inhibit AMP-activated protein kinase (AMPK) Thr172 phosphorylation. IL-6 induces the activation of the Janus kinase/signal transducers and activators of transcription (JAK-STAT) pathway and leads to the inhibition of cytokine signaling 1 and 3 (SOCS1 and SOCS3), thereby reducing insulin receptor sensitivity. Some types of myosteatosis include intermuscular adipose tissue (IMAT), intramuscular fat, and intramyocellular lipid droplets (IMCLs), which contain triglycerides and fatty acid derivatives like DAGs and ceramides [157,158,159,160,161].

Muscle contains various populations of stem cells, including satellite muscle cells and fibro/adipogenic progenitors (FAPs). FAPs are the primary population involved in the development of IMAT related to sarcopenia. With the imbalance caused by inflammation, TNF-α signaling, and TGF-β1, FAP survival and adipogenic differentiation are promoted, leading to the formation of IMAT. Myosteatosis is also associated with oxidative stress and apoptosis due to increased lipid metabolism. As mentioned, one of the main mechanisms for developing muscle insulin resistance is the accumulation of lipid metabolism products, such as DAG, ceramides, and other lipids in IMCL. In obese and insulin-resistant individuals, elevated levels of ceramides can directly cause cellular insulin resistance by inhibiting downstream effects of insulin signaling, such as GLUT4 translocation. Also, DAGs can cause insulin resistance by activating PKCθ, a PKC isoform, leading to the inhibition of IRS1. Another contributing factor to insulin resistance is mitochondrial dysfunction induced by aging, which generates ROS and lipid oxidation, leading to the activation of stress signaling pathways that can inhibit insulin-PI3K-mTOR signaling, such as PKC. Furthermore, impaired insulin signaling exacerbates the inhibition of mitochondrial muscle function, leading to additional lipid accumulation, ROS, and creating a vicious cycle of myosteatosis, lipotoxicity, and muscle insulin resistance. With chronic inflammation and insulin resistance, muscle atrophy is the result. In addition to the direct mechanisms of inflammation that induce atrophy, muscle protein synthesis is also impaired due to the inhibition of insulin signaling, which drives mTORC1 signaling to trigger protein synthesis. In this context, FoxO family transcription factors are also activated by insulin resistance and the presence of ROS. These transcription factors orchestrate a variety of stress responses, such as autophagy during catabolic conditions, but there is interference between ROS action and FoxO, which counter-regulate each other. Atrophy genes (atrogenes), such as atrogin-1 and MuRF-1, are both regulated by FoxO, which in turn activate lysosomal autophagy, contributing to mitochondrial dysfunction and muscle remodeling, resulting in exacerbated muscle atrophy [156,162,163].

Insulin resistance has a key role in pathophysiological mechanisms linking sarcopenia and MAFLD, showing that multiple pathways correlate these conditions, and each condition can exacerbate shared pathways involving inflammation and oxidative stress.

## 6. Relationship Between MAFLD and Sarcopenia

The relationship between MAFLD and sarcopenia can be analyzed from different perspectives, one of which was addressed in a recent study that used data from the Third National Health and Nutrition Examination Survey (NHANES III) and showed that patients with both diseases have a higher risk of mortality than non-carriers. In this study, it is interesting to note that for patients with only one of the disease conditions, it was not possible to define the same correlation, highlighting the associative character between them [7,35,164,165]. A Korean study also reinforces the exposed association and found a result of twice as much mortality risk [166]. Figure 3 shows the crosstalk between the muscle and liver.

Another possible analysis is to inspect the effects of one pathology on another. Patients with MAFLD have a greater loss of muscle mass, increasing the risk of developing sarcopenia. In the pathophysiology of MAFLD, it is possible to observe, as seen above, the important inflammatory state and the insulin resistance, which in this relationship assume the main role due to their ability to increase proteolysis, acting directly on muscle mass loss [167,168]. Similarly, individuals with sarcopenia are associated with a three times higher risk of developing MAFLD, and this makes hepatitis fibrosis even more severe in patients who already have the disease [8,35,37]. In this case, these events are explained by the effects of the increase in insulin resistance from the reduction in muscle mass; after such tissue is a major focus of glucose consumption, which is stimulated by insulin, this soon contributes to a favorable environment for the development of fatty liver, energetic, and inflammatory disorders [37,169].

Finally, the relationship between MAFLD and sarcopenia also presents aspects that can be used for clinical improvement because muscle rehabilitation and dietary therapies in sarcopenia can increase the survival of patients with MAFLD and reduce other complications and comorbidities [170]. It is important to study this relationship because it can influence new specific treatments for both diseases, a scenario that still presents itself as deficient [171].

A multicenter clinical trial conducted in seven hospitals in China demonstrated that intensive lifestyle intervention (ILI), which includes high-protein and low-carbohydrate diets and physical exercise, is able to promote improvements in the framework of MAFLD already ingrained in individuals, and these results are more effective than those obtained through balanced diets with caloric restriction. The patients included in the study were classified as overweight or obese by means of body mass index (BMI) and experienced a reduction in the fat attenuation parameter (FAP), an improvement of lipid metabolism, and increased insulin sensitivity, including a reduction in glycosylated hemoglobin A1c (HbA1C), fasting insulin, and fasting blood glucose [172].

Myosteatosis is a condition characterized by the deposition of intramuscular fat and is related to the loss of muscle function and sarcopenia, in addition to increasing mortality rates in patients with cirrhosis, one of the most serious outcomes of patients with MAFLD. It can be related to MAFLD for its ability to increase insulin resistance and free fatty acids; that is, it can promote pathological factors that contribute to the development of MAFLD. It is worth mentioning that the insulin resistance found in MAFLD can also lead to the development of myosteatosis and consequent sarcopenia, which highlights the relationship between them [173].

A cohort study involving patients with biopsy-confirmed MAFLD and skeletal muscle mass evaluations to characterize sarcopenia found a higher prevalence of sarcopenia in individuals with liver pathologies; sarcopenia was also related to the induction of a high degree of hepatic fibrosis, and this result is maintained regardless of the inflammatory state of the patient, insulin sensitivity, and obesity [174,175]. In addition, another cross-sectional study from Korea also found that such changes promoted by sarcopenia in MAFLD and liver fibrosis cases occur independently of the presence or absence of metabolic syndrome or obesity, which reinforces the correlation described [176].

A cross-sectional study conducted in the USA, composed of elderly patients (60–75 years), showed that there is an important difference between height and weight when readjusting the skeletal muscle index (SMI) of the patients, because when readjusting by height, severe hepatic steatosis is related to a lower risk of sarcopenia, while the weight readjustment implies an increased risk in the same relation, which demarcates the need to continue investigating the definitions that will be used for diagnosis of these diseases in elderly patients [177].

The association between NAFLD and sarcopenia, with a specific focus on gender comparisons, has been the subject of numerous research studies. A recent study was conducted to elucidate this relationship while considering the criteria set by the Asian Working Group for Sarcopenia (AWGS). The study concluded that sarcopenia is associated with a higher prevalence of NAFLD, particularly in men [178].

Recent research has focused on the connection between sarcopenia and non-alcoholic fatty liver disease (NAFLD) in patients with type 2 diabetes (T2DM), particularly examining the association between genders. After collecting and analyzing data, it was found that the prevalence of NAFLD was significantly higher in men with T2DM and sarcopenia compared to those without sarcopenia. However, in women with T2DM, this association was not significant [179].

A study was conducted to explore the relationship between non-alcoholic fatty liver disease (NAFLD/MAFLD) and sarcopenia, independent of visceral adiposity. The study analyzed ultrasonography data from a specific population and found that sarcopenia was strongly linked to both the presence and severity of NAFLD. Additionally, the study discovered that young individuals have a stronger association between these two factors [180].

MAFLD is highly prevalent in patients with chronic obstructive pulmonary disease (COPD) regardless of cardiac comorbidities, metabolic issues, and systemic inflammation. Researchers studied the association of sarcopenia with NAFLD in a population of COPD patients. The study concluded that regardless of age, lung function, and sex, the presence of sarcopenia is linked to a higher risk of NAFLD in COPD patients. Sarcopenic COPD was also associated with a higher percentage of liver fibrosis in patients with NAFLD [181].

A study analyzed anthropometric data to see if the amount of visceral fat and the quantity and quality of muscle mass affect the severity of fibrosis in non-alcoholic fatty liver disease (NAFLD). The study used data from computed tomography (CT) scans and found that there is indeed an association between these factors [182].

It is still being studied whether the low amount of muscle mass and the low quality of muscle strength maintain an independent or joint association with NAFLD. Research results demonstrate that both are independently associated, and when presented in a state of sarcopenia, the risk of NAFLD is higher. Additionally, sarcopenic obesity (SO) increases the risk of NAFLD [183].

A sedentary lifestyle and lack of physical activity are closely linked to obesity and NAFLD. A study utilized data from the National Health and Nutrition Examination Survey (NHANES) to examine mortality rates over several years. The study concluded that a lack of physical activity is associated with the development of sarcopenia, and the presence of sarcopenia is linked to increased mortality in NAFLD patients [184].

A cross-sectional analysis that investigated the relationship between the pathophysiological mechanisms of sarcopenia and NAFLD, through data from the Fifth Korea National Health and Nutrition Examination Survey, correlated the skeletal muscle index (SMI), calculated by dividing the appendicular skeletal muscle mass by total weight and the presence of NAFLD and defined as a fatty liver index (FLI) > 60 in the absence of other liver diseases. It concluded that a reduced SMI is associated with the risk of NAFLD, independent of other widely recognized metabolic risk factors in both sexes [185].

Research examined the relationship between sarcopenia and NAFLD in non-obese children and adolescents through a retrospective analysis of medical records. It utilized the skeletal muscle mass (SMM), the ratio of skeletal muscle to body fat, and the presence of NAFLD via abdominal ultrasound. It concluded that NAFLD is significantly associated with low SMM and that increasing SMM may be suggested as one of the therapeutic approaches for NAFLD in children and adolescents without obesity [186]. Another study analyzed this relationship in children and adolescents, albeit with obesity and overweight as a criterion. It was confirmed that the presence of abdominal obesity, lipid profile alterations, insulin resistance, NAFLD, metabolic syndrome, and NASH was significantly higher in patients with reduced relative muscle mass, and it was concluded that measuring muscle mass may be an approach for identifying metabolic risk in obese children and adolescents as a way to prevent NAFLD and its progression [187].

An investigation of data from the Korean National Health and Nutrition Examination Surveys (2008–2011) on the relationship between sarcopenia and significant liver fibrosis in individuals with NAFLD showed that among individuals diagnosed with NAFLD, sarcopenia was identified in 12.2% of cases and showed a significant association with significant liver fibrosis, independent of obesity and insulin resistance [188].

An analysis indicated that sarcopenia, frequently observed in patients with terminal liver diseases, is associated with worse prognoses, complications, and mortality. The presence of this condition was assessed in patients on the waiting list for liver transplantation, specifically in cases of NASH and alcoholic liver disease (ALD). In cases of NASH, sarcopenia did not show an association with complications, but the frailty score indicated a significant increase in the duration of hospitalizations [189].

A study that investigated the value of the skeletal-to-visceral ratio (SVR) in T2DM patients to predict the presence of MAFLD revealed an increased prevalence of NAFLD in patients with a reduced skeletal-to-visceral ratio (SVR) and that patients with T2DM who have low SVR levels are linked to worse prognoses and complications related to NAFLD. It is suggested that SVR may be a good index for predicting the risk of hepatic steatosis in T2DM [190].

The different pathways that correlate MAFLD and sarcopenia are related to metabolic alterations, oxidative stress, and inflammatory processes. Therefore, it is necessary to control these steps in order to combat the increased incidence of these diseases.

## 7. The Role of Myokines and Hepatokines in Sarcopenia and MAFLD

It is well accepted that skeletal muscles, in addition to their primary function, are involved in various activities, such as producing and releasing cytokines. They also play a role in influencing neighboring cells (paracrine action), themselves (autocrine action), and other tissues (hormonal action). Dysfunctions in myokine production may contribute to the development of metabolic diseases such as type 2 diabetes mellitus, obesity, sarcopenia, and sarcopenic obesity. The aging process can also impact the secretion of nearly all myokines, including insulin-like growth factor 1 (IGF-1), IL-5, L-β-aminoisobutyric acid (BAIBA), apelin, secreted proteins acidic and rich in cysteine (SPARCs), irisin, sesterin, and decorin, while the levels of myostatin secretion increase, leading to a direct inhibitory effect on muscle growth, which is closely related to sarcopenia. Regular physical exercise, especially resistance training, can partially counteract these effects by promoting muscle hypertrophy, and these results are influenced by myokines produced in the muscles during exercise. Recent research suggests that myokines can also serve as therapeutic options and biomarkers for sarcopenia and sarcopenic obesity [191,192,193,194,195]. Table 1 summarizes some hepatokines and myokines and their role in MAFLD and sarcopenia.

Some of the main myokines involved in the pathophysiology of sarcopenia and their respective functions are the following: (1) IGF-1 is an essential growth factor that plays a role in controlling anabolic and catabolic processes, as well as muscle growth and regeneration; (2) myostatin or growth differentiation factor 8 (GDF-8) work as a regulator of muscle mass development and growth by inhibiting the skeletal muscle of synthesizing proteins; (3) irisin is a myokine released under exercise stimulation and is related with adipose tissue loss and insulin sensitivity; (4) Meteorin-like factor (metrnl) is a myokine able to turn the white adipose tissue into brown adipose tissue and is associated with the decrease in insulin resistance; (5) Brain-Derived Neurotrophic Factor (BDNF) is categorized as a neurotrophin that impacts myogenesis and the activation of satellite cells in skeletal muscle; (6) fibroblast growth factor 21 (FGF21) is known to be essential in the regulation of metabolic activities, and both myocytes and adipocytes can be a source; (7) β-Amino isobutyric acid (BAIBA) is secreted during muscle contraction, acts as an inflammatory suppressor, and is linked with adipose tissue loss; (8) apelin is considered a myokine and adipokine, is released with the influence of exercise, has regenerative and anti-inflammatory action, and is indicated to be a pharmacological agent and biomarker of sarcopenia; (9) decorin is released under the stimulation of muscle contraction and stimulates some muscle growth by blocking myostatin; (10) IL-6 is a myokine prototype liberated into the blood circulation when the muscle contracts and may stimulate anti-inflammatory and pro-inflammatory responses; (11) IL-7 is important to the myogenesis process and may stimulate satellite cells differentiation; (12) IL-15 was newly proposed to have the action of a myokine in the metabolism regulation [140,196,197,198,199,200,201,202].

Current studies indicate that a fatty liver affects the production and release of hepatokines in the bloodstream, unlike a healthy liver. Cytokines from metabolic organs such as the liver, muscle, and fat tissue are crucial in maintaining liver and systemic lipid balance. These organokines can influence the synthesis, breakdown, and transport of lipids. Hepatokines play a vital role in metabolic activity and pathological processes as pro-inflammatory factors from the liver. Understanding the relationship between hepatokines and MAFLD is essential for developing effective treatments for this disease. Additionally, hepatokines can serve as biomarkers of fat excess in the liver and as indicators of the development of hepatic diseases [28,203,204].

Some of the hepatokines most related to MAFLD and their functions are the following: (1) Angiopoietin-like 4 (ANGPTL 4) is secreted during physical exercise; this hepatokine promotes less fat absorption by inhibiting pancreatic lipases, and obese individuals have lower levels, which encourages fat accumulation. (2) Leukocyte Cell-Derived Chemotaxin 2 (LECT2) is an organokine that acts as a chemotactic to neutrophils and is closely linked to oxidative stress and weight gain, encouraging the increase in inflammatory cytokines and damaging insulin signaling; it can also be used to identify hepatic steatosis. (3) Sexual hormone-binding globulin (SHBG) is a carrier protein for sexual steroids. It acts as a protective factor against MAFLD by inhibiting the process of lipogenesis. Those with obesity, type 2 diabetes, and hepatic steatosis have reduced levels of SHBG, which can increase with weight loss and exercise. (4) Chemerin is an adipokine and hepatokine involved in the regulation of lipolysis, glucose uptake, and the differentiation of adipocytes. Its activation is mediated by coagulative and inflammatory proteases. (5) Fetuin-A is considered a hepatokine and an adipokine. Elevated levels of this protein are observed in carriers of inflammatory and metabolic diseases. It induces insulin resistance by inhibiting insulin receptors and their signaling. (6) Fibroblast growth factor 21 (FGF-21) is secreted in the liver, muscle, and adipose tissue. It acts as a regulator of metabolic processes involving lipids and glucose. FGF-21 is considered a new marker of MAFLD and may offer a therapeutic alternative. (7) IL-6 can be considered a myokine, hepatokine, and adipokine. While the IL-6 produced in muscle tissue may have a pro-inflammatory function, that produced in the liver and adipose tissue may have pro-oncologic and pro-inflammatory actions [205,206,207,208,209,210,211,212].

The interplay between myokines and hepatokines is crucial for the muscle–liver interlocution. Different compounds, such as lifestyle and lack of physical exercise, can increase the release of these molecules in order to aggravate MAFLD and sarcopenia.

## 8. Prevention of MAFLD and Sarcopenia

Since MAFLD and sarcopenia share mechanisms in their physiopathology, besides risk factors, as an initial approach, searching for similar prevention and treatment for both pathologies is justified. Lifestyle change is currently the most effective and accepted strategy, including diet, exercise, smoking and alcohol cessation, and weight reduction. Currently, no consensus for pharmacological therapy or dietary supplementation has been established, although some alternatives can be mentioned [213,214,215].

The dietary pattern of individuals with MAFLD is typically characterized by a high consumption of processed and ultra-processed foods, rich in calories, sodium, sugars, and saturated fats and low in fiber and vitamins. This eating behavior favors visceral fat accumulation, liver damage, and intestinal dysbiosis, promoting a pro-inflammatory environment that accelerates liver disease progression [216,217].

Weight loss is an essential aspect in the management of MAFLD and sarcopenic obesity. It is acknowledged that reducing visceral fat, balancing one’s lipid profile, and restoring intestinal microbiota ameliorate this comorbidity, and these goals can be achieved only with the correct diet and physical exercise [218,219,220]. Furthermore, weight loss is directly associated with the improvement or resolution of MAFLD, regardless of the underlying metabolic dysfunction [221]. However, this loss must occur gradually and be sustained to ensure lasting improvements in liver health and prevent sarcopenia, since the loss of muscle mass can aggravate or precipitate MAFLD. The restoration of relative skeletal muscle mass may act as a protective factor in preventing and progressing MAFLD, both in healthy individuals and in patients with sarcopenia [222].

Diets balanced in macronutrients and micronutrients can play a crucial role in slowing or stopping the progression of MAFLD and sarcopenia. A diet rich in whole grains, fruits, vegetables, healthy fats, lean proteins, and specific nutrients such as omega-3 fatty acids, antioxidants, fiber, and bioactive compounds is recommended. In addition, it is essential to control the consumption of salt, sugars, and saturated fats [218,223] since pro-inflammatory diets can increase the risk of developing MAFLD [224]. However, diets are not limited to the sum of isolated nutrients; each dietary pattern involves complex interactions between its components, which can exert synergistic or antagonistic effects on health indicators [225].

As suggested by international guidelines, one’s diet should be individualized and moderately hypocaloric (deficit of 500–800 kcal/day or daily intake of 20–25 kcal/kg of current weight) [226,227]. However, the necessary caloric deficit in this case results in the loss of muscle mass and adipose tissue, creating a challenge in treating cases associated with sarcopenia. It has been shown that to minimize muscle mass loss, a daily intake of 1.5 g/kg/weight of protein is essential, together with physical exercise, another fundamental pillar for the treatment of sarcopenia. Furthermore, physical exercise is also effective in reducing LDL cholesterol levels and insulin resistance, in addition to promoting weight loss, both aerobic and resistance [22,23].

Diets rich in plant and dairy proteins (especially whey) have been associated with greater skeletal muscle mass and reduced sarcopenia in elderly patients affected by liver disease and sarcopenic obesity. Furthermore, gut microbial diversity is influenced by the source of dietary protein and its amino acid composition. A more favorable gut microbiome is associated with the regular intake of plant-based and fermented proteins from cheese and whey [228,229].

A study of 10,036 participants found that high sodium intake, as measured by urinary excretion, was associated with a significantly increased risk of MAFLD and sarcopenia. Twenty-four-hour sodium excretion was a moderately accurate predictor of MAFLD, with a cutoff value of 2.30 g/day of sodium showing good sensitivity. Participants with higher sodium excretion had a 46% increased risk of liver condition and a 49% increased risk of sarcopenia, suggesting that high sodium intake is a significant risk factor for both conditions [230].

Omega-3 polyunsaturated fatty acids have been shown to play an important role in the treatment of metabolic and hepatic disorders involving MAFLD, as they significantly reduce liver enzymes, serum triglycerides, and hepatic fat content, improving the degree of hepatic steatosis [231,232].

Vitamin D is crucial as an antifibrotic, anti-inflammatory, and insulin-sensitizing agent, affecting liver function and immune-metabolic pathways between the intestine, adipose tissue, muscle, and the liver. Its deficiency is related to MAFLD and sarcopenia, with patients with MAFLD having reduced vitamin D levels, which intensifies inflammation and liver fibrosis. Daily vitamin D supplementation (800–1000 IU/day) improves muscle strength and balance in sarcopenia, especially in individuals with serum levels below 25 nmol/L. Therefore, vitamin D may benefit patients with early-stage MAFLD and mild to moderate liver damage and is relevant for those with sarcopenia [233,234].

Although many nutraceuticals have been studied in relation to MAFLD and have already shown some efficacy (silymarin, berberine, and curcumin), none has sufficient evidence to recommend their routine use, making it necessary to conduct more robust studies for these approaches [213,235].

Reduced diversity of the intestinal microbiota has been observed in patients with MAFLD, suggesting a possible link between intestinal dysbiosis and disease development [224]. Randomized clinical trials on the use of prebiotics (fructooligosaccharides, beta-glucan, psyllium husk, inulin, among others), probiotics, and synbiotics in the treatment of MAFLD in adults have consistently reported beneficial effects on liver enzymes. However, the strength of this evidence is reduced by the heterogeneity of treatments, strain dosages, and duration of trials, in addition to the lack of MAFLD diagnosis supported by biopsy, imaging, or the histological assessment of results [236]. Because of this uncertainty, the European Society for Clinical Nutrition and Metabolism (ESPEN) guidelines do not recommend the widespread use of pre-, pro-, or synbiotics for patients with MAFLD [226,231].

However, the caloric deficit required in this case results in the loss of muscle mass among adipose tissue, thus creating a challenge for treatment in cases associated with sarcopenia. It has been shown that to minimize the loss of muscle mass, a daily intake of 1.2 to 1.5 g/kg/weight of protein is essential, alongside physical exercise, another fundamental pillar for the treatment of sarcopenia. Moreover, physical exercise is also effective in reducing low-density lipoprotein cholesterol (LDL-c) levels and insulin resistance and promoting weight loss, both aerobic and endurance [29,220,237,238,239,240,241,242].

Lifestyle changes are essential for the prevention of MAFLD and sarcopenia, as these strategies address shared pathophysiological mechanisms such as insulin resistance, oxidative stress, and chronic inflammation.

## 9. Conclusions

MAFLD and sarcopenia are two distinct conditions but interconnected by a complex network of pathophysiological mechanisms, allowing them to interfere with each other in the pathological process. In addition, the association between these diseases alters the prognosis of patients, increasing morbidity and mortality rates. Considering the mechanisms involved that were described in this study, insulin resistance, skeletal muscle tissue secretion capacity, chronic inflammatory state, and oxidative stress are the main contributors to this relationship.

Patients with sarcopenia are more prone to MAFLD with progression to liver fibrosis. In addition, in a clinical analysis, patients with sarcopenia and MAFLD manifest more severe hepatitis fibrosis when compared to patients with only MAFLD. These patients, with both disorders, also present clinical improvement in the MAFLD when treated for sarcopenia, reinforcing the association between them.

There is an intertwined relationship between muscle tissue, adipose tissue, and the liver since the secretion of myokines, hepatokines, and adipokines exerts pleiotropic, redundant, and even antagonistic effects. In the latter case, the increase in muscle mass accompanied by changes in the muscle secretory profile positively interferes (counters) with hepatic metabolism and the deleterious effects of some adipokines. These findings underscore the importance of early screening for sarcopenia in patients with MAFLD, as this can significantly impact disease prognosis and therapeutic strategies. It is also important to include sarcopenia investigation in the MAFLD protocols. This could help with diagnostic and therapeutic approaches and improve quality of life. Nutritional support, physical activity practice, and adequate pharmacological therapies could help in this whole scenario.

## 10. Future Perspectives

The mechanisms involved in the development of MAFLD and sarcopenia have not yet been fully elucidated, and it is necessary to continue developing rigorous clinical studies in order to explore the new pathophysiological possibilities that the change in nomenclature from NAFLD to MAFLD allows. Therefore, it is necessary to expand the academic knowledge about the clinical management of these patients.

In addition, in order to explore new therapeutic approaches and effective pharmacological and non-pharmacological treatments, it is urgent to establish an international consensus on the definition and diagnosis of sarcopenia. Thus, it is known that many obstacles are still encountered in fully unraveling the deep relationship between these diseases.

The widespread adoption of the MAFLD nomenclature as the unification of international consensus on the definition and diagnosis of sarcopenia is extremely necessary for monitoring and epidemiological studies to better understand this liver–muscle relationship (muscle–liver axis). Studies that study the relationship between sarcopenic obesity and MAFLD should be encouraged since the quality of the muscle interferes with its secretory pattern, which in turn interferes with lipid deposition in the liver, the degree of tissue inflammation, and, consequently, susceptibility to progression to fibrosis and possibly cirrhosis.

Especially in patients with myosteatosis (sarcopenic obesity), it is entirely plausible that some cases of MAFLD would be a consequence of high fat mass (obesity) and low muscle quality and quantity.

In an ideal scenario, in the future, algorithms will be developed that allow for the selection of patients with sarcopenia who should undergo liver evaluation and the selection of patients with MAFLD, in whom it is necessary to evaluate both the quantity and quality of skeletal muscle.

## Figures and Tables

**Figure 1 ijms-26-04673-f001:**
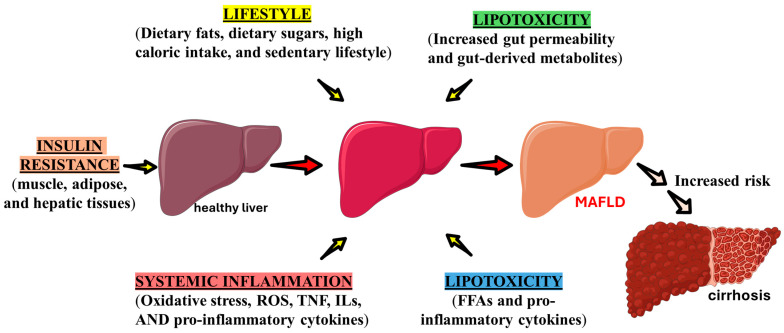
Effects of dysfunctional metabolism that triggers the pathogenesis of MAFLD. FFAs: fatty acids; ILs: interleukins; ROS: reactive oxygen species; TNF: tumor necrosis factor.

**Figure 2 ijms-26-04673-f002:**
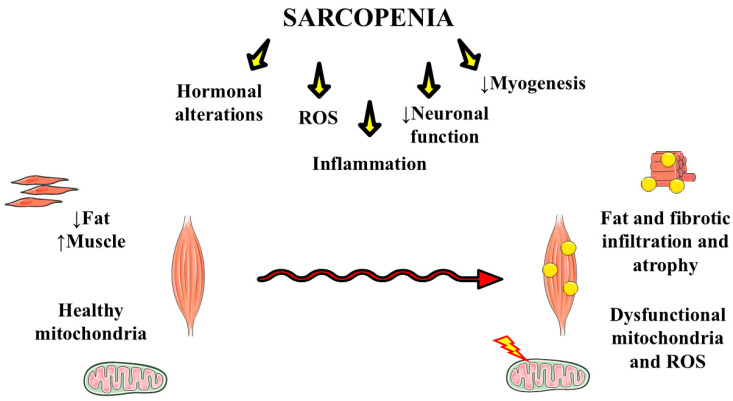
Comparison between a healthy and a sarcopenic muscle, highlighting pathogenic factors. ROS: reactive oxygen species. ↓: reduction.

**Figure 3 ijms-26-04673-f003:**
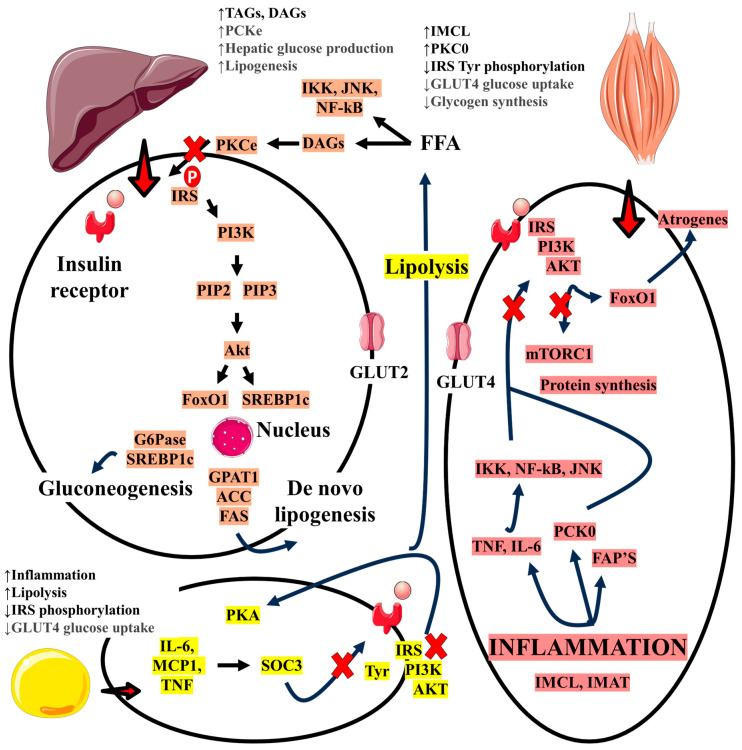
Crosstalk between muscle and liver in sarcopenia and MAFLD conditions. ACC: acetyl-CoA carboxylase; AKT: protein kinase B; DAGs: diacylglycerols; FAPs: fibro/adipogenic progenitors; FAS: fatty acid synthase; FFAs: fatty acids; FoxO1: forkhead box O1; G6Pase: glucose 6-phosphatase; GPAT1: glycerol-3-phosphate acyltransferase; GLUT2: glucose transporter 2; GLUT4: glucose transporter 4; IKK: inhibitor of nuclear factor-κB kinase; IL-6: interleukin-6; IMCLs: intramyocellular lipid droplets; IMAT: intermuscular adipose tissue; INSR; insulin receptor; IRS: insulin receptor substrate; JNK: C-Jun N-Terminal Kinases; MCP1: monocyte chemoattractant protein-1; mTORC1: mechanistic target of rapamycin complex 1; NfkB: nuclear factor κB; PI3K: phosphoinositide 3-kinase; PIP2: phosphatidylinositol 4,5-bisphosphate; PIP3: phosphoinositide 3,4,5 triphosphate; PKA: protein kinase A; PKCε: protein kinase C epsilon; PKCθ: protein kinase C theta; SOCS3: Suppressor of Cytokine Signaling 3; SREBP1c: Sterol regulatory element-binding protein 1c; TAGs: triglycerides; TNF: tumor necrosis factor; Tyr: tyrosine. ↓: decrease; ↑: increase.

**Table 1 ijms-26-04673-t001:** Summary of the role of hepatokines and myokines in MAFLD and sarcopenia.

Myokine	Role in MAFLD	Role in Sarcopenia
IGF-1		Controls the anabolic and catabolic processes, in addition to muscular growth and regeneration.
GDF-8		Regulates the development and growth of muscle mass by inhibiting the protein synthesis of skeletal muscle.
Irisin	Related to the loss of adipose tissue and improvement in insulin sensitivity after physical exercise.	
METRNL	Able to convert white adipose tissue into brown adipose tissue, decreasing insulin resistance.	
BDNF		Acts on myogenesis and activation of satellite cells in the skeletal muscle.
FGF-21	Regulates the metabolic activity.	Regulates the metabolic activity.
BAIBA	Suppresses inflammation, which is linked to the loss of adipose tissue.	Suppresses inflammation.
Apelin		It has regenerative and anti-inflammatory action and is indicated as a pharmacological agent and biomarker of sarcopenia.
Decorin		Stimulates muscle growth by inhibiting myostatin.
IL-6	Can stimulate both anti-inflammatory and pro-inflammatory responses.	Can stimulate both anti-inflammatory and pro-inflammatory responses.
IL-7		Assists in the process of myogenesis and also stimulates the differentiation of satellite cells.
IL-15	Participates in the regulation of metabolism.	Participates in the regulation of metabolism.
**Hepatokine**	**Role in MAFLD**	**Role in sarcopenia**
ANGPTL 4	Promotes lower fat absorption by inhibiting pancreatic lipases, including obese individuals who have lower levels, which encourages the accumulation of fat.	
LECT2	Acts as a chemotactic to neutrophils and is closely linked to oxidative stress and weight gain, encouraging the increase in inflammatory cytokines and damaging insulin signaling.	Acts as a chemotactic to neutrophils and is closely linked to oxidative stress and weight gain, encouraging the increase in inflammatory cytokines and damaging insulin signaling.
SHBG	Acts as a protective factor against MAFLD by inhibiting the lipogenesis process.	
Chemerin	Participates in the regulation of lipolysis, glucose absorption, and adipocyte differentiation.	
Fetuin-A	Induces insulin resistance by inhibiting insulin receptors and their signaling.	Induces insulin resistance by inhibiting insulin receptors and their signaling.
FGF-21	Acts as a regulator of metabolic processes involving lipids and glucose. FGF-21 is considered a new marker of MAFLD.	
IL-6	It may have pro-oncologic and pro-inflammatory actions.	

Abbreviations: ANGPTL 4: Angiopoietin-like 4; BAIBA: β-Amino isobutyric acid; FGF-21: BDNF: Brain-Derived Neurotrophic Factor; fibroblast growth factor 21; GDF-8: myostatin or growth differential factor 8; IL: interleukin; LECT2: Leukocyte Cell-Derived Chemotaxin 2; SHBG: sexual hormone-binding globulin; METRNL: Meteorin-like factor.

## Data Availability

No new data were created or analyzed in this study. Data sharing is not applicable to this article.

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
