# Peer review of "Unraveling the Metabolic Pathways Between Metabolic-Associated Fatty Liver Disease (MAFLD) and Sarcopenia"

_ijms, 2025, doi:10.3390/ijms26104673_

Round 1
Reviewer 1 Report
Comments and Suggestions for Authors
The review article by Losasso ME, et al, is interesting. It deals in detail with the current issue of muscle involvement along with the liver in cases with metabolic syndrome (MS). The authors vividly describe how insulin resistance is involved not only with fatty infiltration and inflammation of the liver, as we used to think until recently, but also with corresponding phenomena from the muscular system. In particular, in elderly people with poor nutrition and immobility, fatty muscle infiltration can gradually lead to a decrease in mass, muscle weakness and sarcopenia. The article is well written in good English and with extensive modern bibliography. It emphasizes the interaction of MASLD and sarcopenia and makes a detailed reference to the individual multiple biological mechanisms that characterize both conditions. However, if the article is primarily addressed, not to biologists, but to a readership of clinicians, the detailed biological mechanisms should, in my opinion, be limited and greater emphasis should be placed on the clinical aspect of this dual problem, namely the frequency of coexistence of hepatic and muscular steatosis; the methods of diagnosing sarcopenia; the progression, the systemic effects, and the efficacy of current pharmacological treatment of MAFLD and its effect on sarcopenia.
I have suggested several changes hoping to improve this interesting article.
General Comments:
- Long article (17 pages of text) with too many references (239). The text should be limited to no more than 12 pages and up to 150 references at most. Long reviews are tiring and not readable.
- Make a conclusion summary at the end of sections 2-7.
- The possible effect of genetic polymorphism of PNPLA3 and TM6SF2 and possibly other genetic factors, must be considered in this review.
- A few typing errors in the text need correction.
Major Comments:
- (Figure 1, line 78): The Figure gives the impression that all NAFLD cases lead to cirrhosis of the liver. This is not correct. Please, consider change.
- (Lines 261-263, 319, 334, 535-539): Please give appropriate references.
- (Live 346, Table 1, 585): Please, explain any abbreviation when first appearing in the text.
- (Live 384): This section must be shortened.
- (Line 746): The long paragraph "Author Contribution" needs to be rewritten. I believe that in each of the 9 parameters mentioned, it is not conceivable that all authors have participated and contributed equally to all parameters. Please assign each author his (her) own specific role in the creation of this review article.
Minor Comments:
- (Live 34): Better use the word "production" instead of "secretiom".
- (Line 81): Please consider changing the phrase as follows: "Sarcopenia is a condition associated by deficient muscle mass and decreased physical performance".
- (Line 90): “sarcopenia”.
- (Line 91): "...implies a lousy prognosis": Please, change the phrase..
- (Line 104): "...insulin resistance".
- (Line 105-106): “It is noticed in patients with lower muscle strength levels of inflammatory makers above average”: Please check this sentence. Something is missing.
- (Line 332): "Peroxisome": No need for capitalization.
- (Line 347): "binding" or "attached to"?
- (Line 349): nf-κB instead?
- (Lines 354-255): “Some types of myosteatosis include intermuscular adipose tissue (IMAT), intramuscular fat…”: One of the 2 phrases is enough. Not both.
- (Line 389): Pathologies: Please consider instead "disease conditions".
- (Line 393): Some abbreviations are not explained in the footnote of the figure. Please correct it.
- Line 439"biopsy-confirmed".
Author Response
Comments and Suggestions for Authors
The review article by Losasso ME, et al, is interesting. It deals in detail with the current issue of muscle involvement along with the liver in cases with metabolic syndrome (MS). The authors vividly describe how insulin resistance is involved not only with fatty infiltration and inflammation of the liver, as we used to think until recently, but also with corresponding phenomena from the muscular system. In particular, in elderly people with poor nutrition and immobility, fatty muscle infiltration can gradually lead to a decrease in mass, muscle weakness and sarcopenia. The article is well written in good English and with extensive modern bibliography. It emphasizes the interaction of MASLD and sarcopenia and makes a detailed reference to the individual multiple biological mechanisms that characterize both conditions. However, if the article is primarily addressed, not to biologists, but to a readership of clinicians, the detailed biological mechanisms should, in my opinion, be limited and greater emphasis should be placed on the clinical aspect of this dual problem, namely the frequency of coexistence of hepatic and muscular steatosis; the methods of diagnosing sarcopenia; the progression, the systemic effects, and the efficacy of current pharmacological treatment of MAFLD and its effect on sarcopenia.
I have suggested several changes hoping to improve this interesting article.
Response: Dear reviewer, thank you for your time revising this manuscript and your kind comments. We improved it according to your suggestions. Please see below a point-by-point response.
General Comments:
- Long article (17 pages of text) with too many references (239). The text should be limited to no more than 12 pages and up to 150 references at most. Long reviews are tiring and not readable.
Response: Dear reviewer, thank you very much for this comment. Our article addresses the interrelationship between Fatty Liver Disease Associated with Metabolic Dysfunction and sarcopenia, two conditions that share common pathophysiological mechanisms, such as insulin resistance, chronic inflammation, and hormonal changes. Recent studies highlight that the coexistence of these conditions is associated with a significant increase in mortality and progression of liver fibrosis. Due to the breadth and complexity of this correlation, we chose to make the reader aware of all the characteristics surrounding these conditions. Therefore, the review is detailed and consequently extensive. We would like to maintain this broad discussion so that not only clinicians, but also other researchers in the health field have, in a single text, the possibility of understanding the connection between MAFLD and sarcopenia.
- Make a conclusion summary at the end of sections 2-7.
Response: Dear reviewer, thank you for this comment. We included your suggestion. Please see page 6, lines 235-237; page 7, lines 315-318; page 9, lines 404-407; page 13, lines 550-554; page 15, lines 628-631; page 16, lines 717-720.
- The possible effect of genetic polymorphism of PNPLA3 and TM6SF2 and possibly other genetic factors, must be considered in this review.
Response: Dear reviewer, thank you for suggesting the inclusion of genetic polymorphism of PNPLA3 and TM6SF2. Please see this inclusion on pages 4-5, lines 165-169.
- A few typing errors in the text need correction.
Response: Dear reviewer, thank you for this observation. We revised all the text.
Major Comments:
- (Figure 1, line 78): The Figure gives the impression that all NAFLD cases lead to cirrhosis of the liver. This is not correct. Please, consider change.
Response: Dear reviewer, thank you for this observation. We improved this figure. Please see on page 2.
- (Lines 261-263, 319, 334, 535-539): Please give appropriate references.
Response: Dear Doctor, the text placed in these lines was based on the cited authors. In any case, I have included other references to improve the association. Please see references og Bagnato et al, 2025 (reference 131), and Sun et al, 2025 (reference 153); please also see reference 189.
- (Live 346, Table 1, 585): Please, explain any abbreviation when first appearing in the text.
Response: Dear doctor, thank you. We included the meaning the first time they appear in the text. Please see pages 13-14, lines 566-569.
- (Live 384): This section must be shortened.
Response: Dear reviewer, thank you. We would like to keep this detail so that the reader has a broad view of the subject.
- (Line 746): The long paragraph "Author Contribution" needs to be rewritten. I believe that in each of the 9 parameters mentioned, it is not conceivable that all authors have participated and contributed equally to all parameters. Please assign each author his (her) own specific role in the creation of this review article.
Response: Dear reviewer, thank you for this comment. We corrected this mistake. Please see lines 773-781.
Minor Comments:
- (Live 34): Better use the word "production" instead of "secretiom".
Response: corrected.
- (Line 81): Please consider changing the phrase as follows: "Sarcopenia is a condition associated by deficient muscle mass and decreased physical performance".
Response: corrected.
- (Line 90): “sarcopenia”.
Response: corrected.
- (Line 91): "...implies a lousy prognosis": Please, change the phrase..
Response: corrected.
- (Line 104): "...insulin resistance".
Response: corrected.
- (Line 105-106): “It is noticed in patients with lower muscle strength levels of inflammatory makers above average”: Please check this sentence. Something is missing.
Response: corrected.
- (Line 332): "Peroxisome": No need for capitalization.
Response: corrected.
- (Line 347): "binding" or "attached to"?
Response: corrected.
- (Line 349): nf-κB instead?
Response: corrected.
- (Lines 354-255): “Some types of myosteatosis include intermuscular adipose tissue (IMAT), intramuscular fat…”: One of the 2 phrases is enough. Not both.
Response: corrected.
- (Line 389): Pathologies: Please consider instead "disease conditions".
Response: corrected.
- (Line 393): Some abbreviations are not explained in the footnote of the figure. Please correct it.
Response: corrected.
- Line 439"biopsy-confirmed".
Response: corrected.
Dear Doctor,
Thank you for your valuable feedback, which has helped us significantly improve the clarity and scientific rigor of our manuscript.
Reviewer 2 Report
Comments and Suggestions for Authors
Manuscript Title: Unraveling the Metabolic Pathways Between Metabolic-Associated Fatty Liver Disease (MAFLD) and Sarcopenia
General Comments:
This review addresses an important and increasingly relevant topic: the relationship between MAFLD and sarcopenia. The manuscript provides a broad overview of potential shared mechanisms, including insulin resistance, chronic inflammation, oxidative stress, and endocrine interactions. The authors also highlight therapeutic strategies and the potential bidirectional impact of these conditions. However, the manuscript requires significant revision to meet the standards of a rigorous scientific review article.
Major Points:
- The manuscript does not describe how studies were identified, selected, or evaluated. A review article should clearly present its methodology to ensure transparency and reproducibility (e.g., databases used, search terms, inclusion/exclusion criteria, and time frame).
- The manuscript summarizes numerous studies without evaluating their quality or discussing limitations. Key mechanistic and clinical associations are presented descriptively, without indicating the strength of evidence, type of study design, or potential biases.
- Core concepts such as insulin resistance, oxidative stress, and inflammatory cytokines (e.g., TNF-α, IL-6) are repeated across multiple sections with minimal added insight. The structure would benefit from consolidation and clearer organization to improve clarity and flow.
- The article sometimes uses subjective or speculative language (“we believe,” “we consider”), which is inconsistent with a neutral scientific tone. Furthermore, it lacks a clear focus—oscillating between mechanistic exposition, clinical narrative, and advocacy without a coherent framework.
- While lifestyle and therapeutic recommendations are mentioned, there is no structured summary or evidence-based evaluation of interventions. This reduces the practical relevance of the review. A dedicated section or table with therapeutic implications, mechanisms, and evidence levels is recommended.
- Although the manuscript occasionally mentions differences between males and females or elderly individuals, these aspects are not critically explored. Given their clinical importance in both MAFLD and sarcopenia, this is a missed opportunity.
Minor Points:
- Several grammatical and syntactical issues need correction.
- Figures are helpful but should be better integrated and referenced in the text.
- The conclusion reiterates earlier content without offering new synthesis or forward-looking insights.
Author Response
Comments and Suggestions for Authors
Manuscript Title: Unraveling the Metabolic Pathways Between Metabolic-Associated Fatty Liver Disease (MAFLD) and Sarcopenia
General Comments:
This review addresses an important and increasingly relevant topic: the relationship between MAFLD and sarcopenia. The manuscript provides a broad overview of potential shared mechanisms, including insulin resistance, chronic inflammation, oxidative stress, and endocrine interactions. The authors also highlight therapeutic strategies and the potential bidirectional impact of these conditions. However, the manuscript requires significant revision to meet the standards of a rigorous scientific review article.
Response: Dear reviewer, thank you for your time revising this manuscript and your kind comments. We improved it according to your suggestions. Please see below a point-by-point response for each comment.
Major Points:
- The manuscript does not describe how studies were identified, selected, or evaluated. A review article should clearly present its methodology to ensure transparency and reproducibility (e.g., databases used, search terms, inclusion/exclusion criteria, and time frame).
Response: Dear reviewer, thank you for this suggestion. As our review is not systematic, we included a short section showing the search strategies. Please see page 4, lines 142-150.
- The manuscript summarizes numerous studies without evaluating their quality or discussing limitations. Key mechanistic and clinical associations are presented descriptively, without indicating the strength of evidence, type of study design, or potential biases.
Response: Dear reviewer, thank you for this observation. We did not include all this information because this is a narrative review. Normally, systematic reviews require the inclusion of the power of evidence or a description of biases. The aim of this review was to show a big scenario showing the relationship between MAFLD and sarcopenia, and not to evaluate each clinical trial found in the databases. This approach allows for a broader discussion of emerging mechanisms and potential clinical implications, which may not yet be fully understood. Thank you for the opportunity to clarify our approach.
- Core concepts such as insulin resistance, oxidative stress, and inflammatory cytokines (e.g., TNF-α, IL-6) are repeated across multiple sections with minimal added insight. The structure would benefit from consolidation and clearer organization to improve clarity and flow.
Response: Dear reviewer, thank you for this comment. These terms are mentioned several times throughout the text because they are part of the pathophysiological mechanisms of the conditions studied and their multiple actions. We believe that this approach is important to accurately highlight so many interactions that interconnect in sarcopenia and MAFLD. However, we revised the manuscript so that this repetition does not compromise the clarity of the interconnections.
- The article sometimes uses subjective or speculative language (“we believe,” “we consider”), which is inconsistent with a neutral scientific tone. Furthermore, it lacks a clear focus—oscillating between mechanistic exposition, clinical narrative, and advocacy without a coherent framework.
Response: This is also a nice suggestion. We removed “we” from the entire text. Furthermore, you are correct in saying that there is a need to maintain a coherent narrative throughout the manuscript. However, our intention was to provide a comprehensive discussion that paints the full picture of the complexity of the relationship between MAFLD and sarcopenia. Given this complexity, it was necessary to address both mechanistic insights and clinical implications to provide a complete picture. This approach reflects the interconnected nature of metabolic pathways and clinical outcomes, which cannot be fully understood in isolation. We believe that this framework is essential to bridge the gap between basic science and clinical practice, making the findings more impactful to a diverse audience. However, we carefully revised the manuscript to ensure the presentation remained clear and organized.
- While lifestyle and therapeutic recommendations are mentioned, there is no structured summary or evidence-based evaluation of interventions. This reduces the practical relevance of the review. A dedicated section or table with therapeutic implications, mechanisms, and evidence levels is recommended.
Response: Dear reviewer, thank you for your comment. The therapeutic aspects were only mentioned because the aim of our study was not to explore the intervention. For these reasons, our focus was based on oxidative stress and inflammation, pathways that interconnect MAFLD and sarcopenia.
- Although the manuscript occasionally mentions differences between males and females or elderly individuals, these aspects are not critically explored. Given their clinical importance in both MAFLD and sarcopenia, this is a missed opportunity.
Response: Dear reviewer, this is an interesting observation. However, our aim was to show the metabolic pathways involved in both conditions (MAFLD and sarcopenia) and not explore the gender aspects. This will be explored in another paper.
Minor Points:
- Several grammatical and syntactical issues need correction.
- Response: Dear reviewer, thank you for this observation. We revised the entire manuscript.
- Figures are helpful but should be better integrated and referenced in the text.
Response: Dear Doctor, thank you for this comment. All figures are referenced in the text.
- The conclusion reiterates earlier content without offering new synthesis or forward-looking insights.
Response: Dear reviewer, thanks for your suggestion. We improved the conclusion. Please see on page 17, lines 722-745
Dear Doctor,
Thank you for your valuable feedback, which has helped us significantly improve the clarity and scientific rigor of our manuscript.
Round 2
Reviewer 1 Report
Comments and Suggestions for Authors
No comments
Reviewer 2 Report
Comments and Suggestions for Authors
None